# Assessing the Risk of Seasonal Effects of *Campylobacter* Contaminated Broiler Meat Prepared In-Home in the United States

**DOI:** 10.3390/foods12132559

**Published:** 2023-06-30

**Authors:** Xinran Xu, Michael J. Rothrock, Govindaraj Dev Kumar, Abhinav Mishra

**Affiliations:** 1Department of Food Science and Technology, College of Agricultural & Environmental Science, University of Georgia, 100 Cedar St., Athens, GA 30602, USA; xinran.xu@uga.edu; 2Egg Safety and Quality Research Unit, U.S. National Poultry Research Center, Agricultural Research Service, United States Department of Agriculture, Athens, GA 30605, USA; michael.rothrock@usda.gov; 3Center for Food Safety, University of Georgia, Griffin, GA 30223, USA; goraj@uga.edu

**Keywords:** *Campylobacter*, risk assessment, food safety, seasonal effect, chicken meat

## Abstract

Campylobacter has consistently posed a food safety issue in broiler meat. This study aimed to create a quantitative microbial risk assessment model from retail to consumption, designed to evaluate the seasonal risk of campylobacteriosis associated with broiler meat consumption in the United States. To achieve this, data was gathered to build distributions that would enable us to predict the growth of Campylobacter during various stages such as retail storage, transit, and home storage. The model also included potential fluctuations in concentration during food preparation and potential cross-contamination scenarios. A Monte Carlo simulation with 100,000 iterations was used to estimate the risk of infection per serving and the number of infections in the United States by season. In the summer, chicken meat was estimated to have a median risk of infection per serving of 9.22 × 10^−7^ and cause an average of about 27,058,680 infections. During the winter months, the median risk of infection per serving was estimated to be 4.06 × 10^−7^ and cause an average of about 12,085,638 infections. The risk assessment model provides information about the risk of broiler meat to public health by season. These results will help understand the most important steps to reduce the food safety risks from contaminated chicken products.

## 1. Introduction

Campylobacteriosis, a foodborne illness caused by bacteria of the genus *Campylobacter*, poses a significant global health burden. This bacterial infection is one of the leading causes of gastroenteritis worldwide, with millions of people affected annually [1] Affecting both developing and developed nations, the disease is primarily linked to the consumption of contaminated poultry, unpasteurized milk, and untreated water. In developed countries, campylobacteriosis is often a sporadic, rather than epidemic, issue, with cases peaking during the warmer months [2]. Conversely, in developing countries, *Campylobacter* infections are often endemic and more frequently affect children under the age of five [3].

In the chicken sector, *Campylobacter* spp. present a persistent food safety risk. According to the Centers for Disease Control and Prevention (CDC), there were 96 campylobacteriosis outbreaks linked to the consumption of chicken in the United States between 1998 and 2020, resulting in 856 illnesses [4]. *Campylobacter* has been demonstrated to reside in the gastrointestinal tract of broilers, which may explain why the bacterium is so commonly connected with poultry-related campylobacteriosis [5]. A study revealed the prevalence of *Campylobacter* spp. in broilers from North Lebanon, with high infection rates in both open rearing system (92%) and closed rearing system (85%) [6]. *Campylobacter* may infect broilers and chicken carcasses at any stage of the broiler supply chain. Examples of preharvest contamination include feed, other farm animals, biosecurity hazards (wildlife species), potable water, soil, insects, farm equipment, personnel, visitors, and farm vehicles [7]. Postharvest contamination is caused by fecal contamination of feathers and skin during transit, fecal material leaks during evisceration, and contact with contaminated equipment and water [8].

Multiple studies have shown substantial seasonal trends in the incidence of *Campylobacter* in the environment and at different levels of the food chain between live animals and human sickness. Stern [9] showed that *Campylobacter* concentrations in the United States were lower in the autumn and spring than in the summer and winter. Willis and Murray [10] observed that from May to October in the United States, the incidence of *Campylobacter* is high (86.7% to 96.7%) based on a one-year study of carcass samples. In addition, research from France, the United Kingdom, and several other nations indicate a greater *Campylobacter* frequency in warm months than in cold ones [11,12,13]. A study carried out in Lebanon also reported the highest *Campylobacter* infection rates in summer (31.6%) [14]. Several studies have also linked human campylobacteriosis incidences to the hottest months of the year [15,16].

Since the late 1990s, when Willis and Murray [10] utilized quantitative microbial risk assessment (QMRA) to estimate the risk of salmonellosis due to the consumption of liquid eggs, QMRA has been a commonly used method in the food industry to evaluate the risk of microbiological hazards to food consumers [17,18]. Quantitative microbial risk assessment (QMRA) is a technique for estimating human health risks based on dose–response (DR) models for particular (reference) pathogens and exposure scenario evaluations [19]. The process consists primarily of determining the concentration of reference pathogens at the points of environmental exposure, typically by estimating the sources and modeling pathogen fate and transport to the points of human exposure; this concentration is then combined with ingestion volume to calculate the dose. Dogan et al. [17] quantified the risk of *Campylobacter* during processing in the United States, Lindqvist and Lindblad [20] summarized the risk of *Campylobacter* during the handling of raw chicken in Sweden, and Hartnett et al. [21] developed a model to assess the risk of *Campylobacter* at the time of slaughter in the United Kingdom. The purpose of this research was to develop a retail-to-consumption QMRA model that could be used to evaluate the seasonal impact of the yearly illnesses induced by the consumption of broiler meat processed at home in the United States.

## 2. Materials and Methods

### 2.1. QMRA Overview

A process flow from the moment that broiler meat is packed till its consumption by customers has been devised (Figure 1). The flow illustrates the human consumption of a portion of meat cooked from a retail-purchased package of chicken. The flow includes retail storage; delivery to the consumer’s house; and storage, preparation, and consumption at the consumer’s residence. A search of the scientific literature was conducted to discover distributions that might be used to explain characteristics in each of these domains, as well as the growth and inactivation kinetics of *Campylobacter* at different temperatures. Table 1 presents the variables used in the QMRA model.

### 2.2. Campylobacter Growth Kinetics

To capture the growth behavior of *Campylobacter* at the vast range of temperatures it may experience along the retail-to-consumption chain, a thorough knowledge of *Campylobacter* growth rates on broiler meat is required. Primary growth data were obtained fromBlankenship [22], Nicorescu and Crivineanu [23], and Solow et al. [24]. Only a limited number of studies were found investigating the growth of *Campylobacter* in chicken meat under various temperatures. For each research, primary growth data were extracted, and the three-phase linear model was fitted to the growth data in order to calculate the specific growth rate (k) using the following equations [52]:(1)$$y_{t}=y_{0}\;\mathrm{for}\;t\leq t_{lag}$$
(2)$$y_{t}=y_{0}+k\left(t-t_{lag}\right)\;\mathrm{for}\;t_{lag}<t<t_{max}$$
(3)$$y_{t}=y_{max}\;\mathrm{for}\;t\geq t_{max}$$
where $y_{t}$ is the population of bacteria at time t (log CFU/g), $y_{0}$ is the initial population of bacteria (log CFU/g), $y_{max}$ is the maximum population of bacteria supported by the environment (log CFU/g), k is the specific growth rate (log CFU/h), t is the elapsed time (h), $t_{lag}$ is the lag time (h), and $t_{max}$ is the time when $y_{max}$ is reached (h). When there were only two phases in the growth data, a biphasic model was fitted, with the phases consisting of either a lag phase and exponential phase, or an exponential phase and stationary phase. Primary models were fitted using the United States Department of Agriculture (USDA) Integrated Pathogen Modeling Program (IPMP; Version 2013) [53].

After estimating k from the primary data, the Ratkowsky model was applied to the growth rates as described by the following equation [54]:(4)$$\sqrt{k}=b\;\left(T-T_{\min}\right)$$
where *T* is the temperature (°C), $T_{\min}$ is the theoretical minimum temperature for growth (°C), and b is a growth constant. It has been shown that the Ratkowsky model should be used for temperatures between the lowest and optimal growth temperatures of an organism; hence, only growth rates from temperatures between 37 and 42 ℃ were employed. Hazeleger et al. [25] and Park [26] showed that *Campylobacter* required a minimum growth temperature of 31 °C. Therefore, if simulated temperatures in the QMRA were below 31 °C, a growth rate of zero was used to indicate no growth. The MATLAB Curve Fitting Toolbox (Version R2019b; Mathworks, Natick, MA, USA) was used to conduct secondary modeling, and estimates for b and $T_{\min}$ were derived. Because of lacking growth data for *Campylobacter* on chicken meat, single estimated values were used in QMRA model.

### 2.3. Product Temperature Change

Newton’s law of heating has been used to explain the change in temperature of a food product when it enters a warmer ambient environment as a function of the product’s initial temperature, the ambient temperature, and the amount of time the product spends in the ambient temperature [55]. It can be described by the following equation:(5)$$T=T_{a}-\left(T_{a}-T_{0}\right)e^{-Bt}$$
where T is the final product temperature (°C), $T_{a}$ is the ambient temperature (°C), $T_{0}$ is the starting product temperature (°C), t is the time in ambient temperature (h), and B is a constant (h^−1^). Using Equation (5), Equation (4) is rewritten to describe the growth rate of *Campylobacter* when chicken enters a warmer ambient temperature (Equation (6)).
(6)$$\sqrt{\mu }=b\;\left(T_{a}-\left(T_{a}-T_{0}\right)e^{-Bt}-T_{\min}\right)$$

The distribution of B is obtained from Golden and Mishra [27]. Equation (6) was used to predict the growth rate of *Campylobacter* when a shift from cold to warm ambient temperature was anticipated; otherwise, Equation (4) was utilized. Due to @Risk software restrictions, only one growth rate was generated for each iteration of the QMRA model, given the time and temperature experienced at that iteration.

### 2.4. Retail Prevalence

Multiple studies reported monthly prevalence data for *Campylobacter* in chicken meat [22,23,24]. Spring (March, April, and May), summer (June, July, and August), autumn (September, October, and November), and winter (December, January, and February) meteorological seasons are used in this study [56]. By season, monthly prevalence statistics were categorized. Then, 1000 samples were generated using the bootstrapping method. The mean values of each sample collected using the bootstrapping approach was computed. The @RISK software was used to fit distributions to these mean values. The initial concentration of *Campylobacter* was determined based on a 2012 countrywide USDA study, in which chicken parts were examined at the end of the manufacturing line and positive samples were measured in CFU/Ml [29]. Due to the availability of data, it Is vital to mention that the concentration was based on chicken samples gathered after manufacturing and before reaching retail. Therefore, it was hypothesized that neither an increase nor a decrease in *Campylobacter* counts happened during transit from the manufacturing site to the retail location.

### 2.5. Retail Storage

The growth of *Campylobacter* in chicken products were assessed in two parts: retail cold room storage time and temperatures, and display storage time and temperatures [30]. An exponential distribution was fit for cold room storage and display storage time. A normal distribution was fit for cold room storage and display temperature.

### 2.6. Transportation and Home Storage

The monthly average ambient daytime temperatures in the United States’ major cities was obtained fromCR [31]. In order to determine the average ambient daytime temperature per season, distributions were developed. Ecosure [32] gathered data on transportation times and fitted them to a log-logistic distribution. The distribution was trimmed at the lowest and highest observed times to prevent impractical low and high values on either ends. The bacterial growth rate during transportation was approximated using Equation (6), where the ambient temperature and time spent at the ambient temperature were selected from the distribution mentioned above, and the initial temperature was determined by the retail storage temperature. In addition, a Beta general distribution was utilized to estimate the period between customers’ arrival at home and the placement of meat products in the refrigerator [33]. Using the ambient room temperature from Booten et al. [34], the growth of *Campylobacter* was approximated prior to refrigeration.

In the baseline model, scenarios were built based on whether or not a customer elected to freeze the chicken meat they bought. In a study conducted by Mazengia et al. [33], it was discovered that forty percent of respondents froze chicken meat before consuming it. In each model iteration, a Bernoulli distribution with *p* = 0.40 was utilized to determine whether or not a customer froze their chicken meat. If a customer did not freeze their meat, it was presumed that it was cooked immediately after being stored in the refrigerator. According to the same poll, consumers kept chicken meat in the refrigerator for an average of one to seven days before consumption [33]. The time meat was held in the refrigerator prior to cooking or freezing was modeled using a Pareto distribution. If a customer chose to freeze their chicken, the same quantity of refrigerated storage was utilized to replicate the time it would take them to store the chicken in frozen storage. It was anticipated that no growth occurred during frozen storage since realistic freezer temperatures would not support the proliferation of *Campylobacter* [57]. After frozen storage, four thawing methods were considered: refrigeration, running water, microwave, and room temperature. Extracted data from a survey performed by Mazengia et al. [33] were used to assess the probability that a customer would utilize one of these method to defrost frozen food. The USDA recommends all methods other than room-temperature thawing for the safe defrosting of meat [58]. Thawing timings for refrigeration and running water were based on USDA standards, while microwave thawing periods were based on common “defrost” settings (25–30% power) for residential microwaves [38,58]. Growth rates were calculated based on thawing duration and temperatures experienced throughout the different thawing methods. For all preparation techniques, it was expected that customers would immediately cook their chicken after thawing.

### 2.7. Cross-Contamination during Preparation

In the baseline QMRA model, the following cross-contamination scenarios were evaluated: raw chicken to hands, raw chicken to utensils (e.g., cutting boards, knives, etc.), hands to cooked chicken, and contaminated utensils to cooked chicken. A number of studies have provided transfer rate data about *Enterobacter aerogenes* and *Campylobacter* spp. during food preparation [39,40,41]. Although data on the transfer rates from raw chicken to hands and raw chicken to utensils were provided, the transfer rates from unclean hands and utensils to cooked chicken were calculated using lettuce, bread, and cucumber, since these information for chicken are not available. For each study, transfer rates were extracted, and distributions were fitted. We calculated the changes in *Campylobacter* concentration after each phase of handling. Kosa et al. [44] estimated that 88.3 percent of individuals wash their hands after handling raw chicken. Therefore, a Bernoulli distribution was utilized to determine whether a decrease in hand washing should be applied to the hand concentration. Chen et al. [39] provided statistics on the decrease in hand washing. If a person used different tools than those used to handle raw chicken, the transfer rate from utensils to cooked chicken was assumed to be 0%.

### 2.8. Cooking

The chicken meat cooking time and temperature data were collected from Oscar [42] andBruhn [35], respectively. A Pert distribution was used to simulate the process of cooking chicken at home. In this stage, the baseline model examined whether or not the chicken product was undercooked. Undercooking is described as cooking chicken meat below the USDA-recommended temperature of 165 °F (73.9 °C) [58]. According to a study conducted by Bruhn [35] Ecosure [32], 39.9% of chicken products were undercooked. If the chicken was adequately cooked, it was considered that the prepared product had 0 CFU/g *Campylobacter*. If the chicken was undercooked, the D-value was calculated based on the inactivation model provided by van Asselt and Zwietering [43]. As higher temperatures were predicted to result in shorter cooking periods, a correlation coefficient of −0.75 was used to represent the link between cooking time and temperature [27].

### 2.9. Dose–Response Modeling and Risk Characterization

Multiplying the concentration of *Campylobacter* in an eaten serving by the serving size yielded the ingested dosage. For all simulations, a serving size of 85 g was adopted based on the reference quantity commonly eaten per eating occasion for chicken meat (9 CFR 381.412) [59]. As a final result, we intend to estimate the probability of infection and illness resulting from the consumption of chicken meat contaminated with *Campylobacter*. Consequently, the evaluation of exposure is dependent on the dose–response relationship. The most common dose–response relationship for *Campylobacter* is the beta Poisson model for infection probability [45,60]. The probability of infection was determined by the following equation:(7)$$P_{inf}=1-{\left(1+\frac{D}{\beta }\right)}^{-\alpha }$$
where $P_{inf}$ is the probability of infection, D is the ingested dose (CFU), and α and β are the model parameters. Regarding to the probability of infection, Black et al. [46] provided data on the likelihood of disease. On the basis that 29 out of 89 infected persons became ill, it is hypothesized that *P_ill|inf_* = 0.33 is a straightforward model for estimating the likelihood of sickness given an infection. The equation for probability of illness is:(8)$$P_{ill}=P_{inf}\times P_{ill|inf}$$

The risk of infection per serving of chicken was then calculated by multiplying the chance of infection by the seasonally retail prevalence [61].

### 2.10. “What-If” Scenarios

The best- and worst-case alternative scenarios for the basic QMRA model were analyzed, and the predicted total number of campylobacteriosis cases from each scenario was compared to the result from the baseline model. The effect of thawing methods (refrigerator thawing, running water thawing, microwave thawing, and ambient room temperature thawing) was considered by running simulations where only one thawing method was applied. The uncertainty of low, medium, and high *Campylobacter* prevalence (based on season) was considered. In addition, scenarios of hand washing (always wash hands and never wash hands) and cleaning (always use different utensils and never use different utensils) were taken into account for uncertainty analysis. For each thawing method, the temperature and time distribution were obtained from either literature or expert opinion (Table 1).

### 2.11. Risk Modeling

Using @Risk software (Version 7.6.1; Palisade, Ithaca, NY, USA), all distribution fitting, correlation matrix application, and simulations were conducted. Where appropriate, *Campylobacter* concentrations were converted to decimal log_10_ values. All Monte Carlo simulations were conducted with a total of 100,000 iterations with Latin hypercube distribution sampling. To serve as the seed for all simulations, a random number between 1 and 100 (chosen number: 28) was selected at random. Using the RiskSimtable function in @Risk, uncertainty assessments were conducted. The correlation coefficients of Spearman were utilized in the sensitivity analysis to examine the influence of distribution factors on output variables.

## 3. Results and Discussion

### 3.1. Seasonal Effect on Presence of Campylobacter in Chicken

Willis and Murray [10] and Hinton et al. [28] provided data on the prevalence of *Campylobacter* in chicken meat on a monthly basis. The seasonal prevalence was computed using the mean and is shown in Table 2. Similarly, monthly *Campylobacter* concentrations were collected, and the seasonal concentration was computed [62]. *Campylobacter* prevalence was stable throughout the spring, summer, and fall months (0.53 to 0.59) but was lower during the winter months (0.26). (Table 2). With minimal change, *Campylobacter* concentrations were lowest during summer (1.74 log CFU/carcass) and highest in Fall (2.35 log CFU/carcass). Berrang et al. [63] examined *Campylobacter* concentrations in cecal samples obtained from a Georgia processing plant. The data demonstrated a similar prevalence pattern. The prevalence of *Campylobacter* was greater in the warmer months (March to November) (0.53 to 0.64) than in the cooler months (December to January) (0.46). In Alabama, 41 percent of skinless chicken breasts were contaminated with *Campylobacter* [64]. Additionally, seasonal patterns were identified in other nations. There was a significant seasonal pattern in retail chicken meat over the summer and Fall months in Denmark [65] and Wales [15]. Lynch et al. [66] found a considerably higher *Campylobacter* prevalence in chicken ceca samples in July (0.85 against other months in Ireland) compared to other months. Garcia-Sanchez [67] determined that spring and autumn are the most important seasonal variables for *Campylobacter* prevalence on a Spanish farm.

### 3.2. Growth Rates

During the literature search, few data on *Campylobacter* growth on chicken meat (chicken parts, ground chicken, or chicken skin) were found. Consequently, the primary growth rates of *Campylobacter* were determined between 37 and 42 °C. The parameters b (0.04673) and $T_{\min}$ (31.96 °C) were determined by fitting the secondary Ratkowsky model to growth rates, yielding an R^2^ value of 0.603. Due to the restricted amount of accessible data points, point estimates were utilized instead of distribution in the QMRA model. Furthermore, the observed minimum growth temperature (31 °C) of *Campylobacter* was determined and included into the QMRA model [25,26].

### 3.3. Effects of Ambient Temperature

As established by Golden and Mishra [27], temperature variation was taken into account during shipping and thawing. Newton’s law of heating was applied to chicken flesh in order to account for the amount of time it takes for chicken to achieve its ambient temperature when placed in a warmer environment. Newton’s heating constant B had an average value of 2.26 h^−1^ (standard deviation: 0.54 h^−1^). This number helps to estimate the surface temperature of chicken after a certain amount of time at a specified ambient temperature. This is crucial for calculating how much pathogens proliferate during the trip from a store’s refrigerated storage to a consumer’s house, since the product’s temperature often rises during this period [68]. Moreover, according to a study, the temperature of fresh meat left in a vehicle trunk for two hours in the summer (average ambient temperature of 32.6 °C) reached 34.4 °C [69]. During transportation, meat products may readily enter the danger zone for microbial development if exposed to high ambient temperatures. The average travel time from grocery shops to customers’ homes was 1.2 h, whereas the USDA Food Safety and Inspection Service recommended that perishable items be refrigerated within two hours [70]. When the outside temperature exceeds 32.2 °C, perishable items must be placed in the refrigerator within one hour.

### 3.4. Baseline QMRA Model

For the seasonal effect, the risk of infection and illness per serving are shown in Table 3. The mean risk of infection per serving was 1.31 × 10^−3^, 1.57 × 10^−3^, 1.45 × 10^−3^, and 7.01 × 10^−4^ for spring, summer, fall, and winter, respectively. These results reflect the seasonal trend seen in retail *Campylobacter* prevalence, where winter season showed a lower value compared to warmer seasons. To calculate the number of illnesses caused in each season in the United States, the total number of servings for each season was calculated using public data and the reference amount commonly eaten (RACC) per eating occasion for chicken meat of 85 g (9 CFR 381.412) [59]. Due to the absence of information about the minimum and maximum serving sizes, only the RACC value of 85 g was employed in this research. This resulted in an estimated total of 17,247,755,827 seasonal meals of chicken meat prepared at home. The estimated average number of infections and illnesses in spring caused by consuming in-home prepared chicken in the baseline QMRA model were 22,571,609 (median 13,050) and 7,448,639 (median 4307), respectively (Table 4). The cumulative distribution of the number of infections and illnesses by season is shown in Figure 2 and Figure 3. Despite the influence of outlier simulation results on the average, these results offer an estimate for the number of infections throughout the population and serves to demonstrate the uncertainty around the estimate, while the median helps to illustrate the distribution of simulation results.

The baseline QMRA model predicted an average of approximately 86,657,118 cases (median: 50,493) of campylobacteriosis infection annually. The estimated number of *Campylobacter* infections are 2.4 million every year [71], which is lower than our prediction. This may be due to the fact that campylobacteriosis is largely underreported [72]. The predicted mean number of illnesses annually was 28,596,849 (median 16,663) from baseline QMRA model. Between 2009 and 2010, the U.S. National Outbreak Reporting System received reports of 56 confirmed and 13 suspected outbreaks, among which 1550 illnesses and 52 hospitalizations were documented [73]. Furthermore, based on outbreak data from 1998 to 2008, it was projected that 845,024 cases of campylobacteriosis occurred year in the United States, resulting in 8463 hospitalizations and 76 fatalities [74]. From 1996 to 2012, the U.S. Food-Borne Diseases Active Surveillance Network reported an annual incidence of *Campylobacter* infection of 14.3 per 100,000 people [75].

An important aspect of campylobacteriosis case distribution is the considerable seasonality and age-related fluctuation in incidence rates [13,76,77] Poultry is of special relevance to the overall epidemiology of campylobacteriosis since it is often infected and may shed the germs in extremely large numbers [26,78]. Following the slaughtering process, the contamination of poultry meat is common, and several case–control studies have linked the handling or ingestion of chicken meat to human illnesses [76]. The season is often connected with temperature and may also impact campylobacteriosis risk due to seasonal differences in human activity, food supply, or changes in natural ecosystems. Higher temperatures may lead to an increase in the incidence of *Campylobacter* in animal populations or water, or to an increase in temperature abuse during food transit, storage, or handling [65]. Seasonality may have an effect independent of temperature since human activities that facilitate exposure to *Campylobacter* fluctuate with the seasons. Seasonal variations in travel; swimming in untreated water; playground use; and direct contact with cattle, other animals, and flies are all related with higher risks of campylobacteriosis.

### 3.5. Uncertainty Analysis

The numbers of infections and illnesses based on thawing methods are shown in Table 5. Thawing chicken meat in ambient room temperature significantly increases the total number of infections. Due to improper thawing, packaging of meat with other ready-to-eat foods, and poor handling of food contact materials, there was a high risk of cross contamination. Mkhungo et al. [79] reported that 28% of people left their meat product on kitchen counter to thaw. Thawing takes more time than freezing, and when ambient air or running water is used, some parts of the raw meat are exposed to temperatures that are conducive for microbial growth [80]. Additionally, the water that comes out of thawing meat is full of nutrients that could help bacteria grow. It does not seem that the amount of live bacteria present in meat is reduced by either the freezing or thawing process. The process of freezing, on the other hand, causes bacteria to enter a state of dormancy, which effectively puts an end to microbial deterioration. During the thawing process, unfortunately, they recover their activity. As a result, ambient room temperature thawing for meat products raises a huge food safety risk for consumers.

Table 5 summarizes the statistics for annual number of infections calculated during the uncertainty analyses. High *Campylobacter* prevalence showed higher mean number of infections (92,231,575), but the difference with low and medium *Campylobacter* prevalence was not significant. Washing hands after handling raw chicken showed great difference in the number of infections. The median cases of always hand washing were 26,583 compared to that of never wash hands was 11,308,686. Similarly, always using different utensils when cooking chicken products showed the median number of infections of 42,190, whereas the number of infections for never using different utensils was 2,694,612. Our results suggest that the cross-contamination during handling and cooking chicken meat showed more significant impact on the risk of campylobacteriosis than the initial prevalence of chicken products. The exposure assessment reveals that cross-contamination is the primary cause of bacteria exposure via food produced in kitchens [81]. The authors also conclude that cross-contamination appears to be a greater concern than bacterial development, even when products are held at high ambient temperatures. Kusumaningrum [82] examined unwashed surfaces as a cross-contamination factor during the preparation of chicken salad using ready-to-eat (RTE) ingredients. On average, 26% of consumers did not wash surfaces while preparing raw and cooked foods or ready-to-eat foods. Furthermore, cross-contamination from chicken to other ingredients via surface may happen. In addition, Lopez et al. [83] found that using disinfectant wipes on kitchen surfaces during preparation chicken meat could effectively reduce the risk of *Campylobacter* infections.

### 3.6. Sensitivity Analysis

Cross-contamination events (hands wash reduction; whether the hands are washed; and transferring from hands to cooked chicken) were the three most significant QMRA variables for predicting total *Campylobacter* risk of infection per serving, followed by the *Campylobacter* concentration at purchase and transfer rate from raw chicken to hands (Figure 4). As we expected, the frequency of washing hands was the most significant factor in reducing the risk. While this may be seen as a method to lower the risk of illness due to the intake of chicken, another concern that should be addressed is the development of bacterial antimicrobial resistance to compounds contained in antimicrobial soaps, such as triclosan [84]. The third most important risk factor identified in the present model is handling cooked chicken with raw-meat-contaminated hands. In a 2008 study, between 73% and 100% of subjects who claimed to have washed their hands after handling raw chicken were found to have *Campylobacter jejuni* on their hands [85]. Similar outcomes were also observed that even when hands are well cleansed, large amounts of bacteria might remain [86]. Moreover, a recent survey revealed that just 39.6% of customers properly cleansed their hands after handling raw chicken breast [87]. These findings and observations indicate that there is still a significant need for improvement in consumer education on the safety of chicken products. Many of the people cleaned their hands by washing or rinsing them after handling the raw chicken items; however, they did not wash their hands until after they had contaminated other parts of the kitchen by touching things such as spices, utensils, or cooking surfaces. Additionally, Signorini et al. [88] found the frequency of washing cutting board and hands were the second and fourth most important factors in human campylobacteriosis risk in a risk assessment carried out in Argentina. The author also reported that during food preparation, the risk of human campylobacteriosis was 1.47 times greater for those who did not wash their hands.

This QMRA model was developed to represent the existing knowledge and practices of the retail-to-consumer supply chain for broiler meat. Consequently, a number of assumptions were included into the model, and knowledge gaps were found. First, there were few data on the development of *Campylobacter* in chicken meat. More information will assist explain the growth characteristics and behavior of *Campylobacter* on chicken more precisely. It was presumed that customers did not transport meat from the grocery to their homes in a chilled state. It is probable that other items purchased with chicken meat might influence the temperature of the chicken during transportation, however no information is available to address this issue. In addition, information on storage and display times in U.S. grocery shops may be required. Next, assumptions have to be established about thawing timeframes for each of the evaluated thawing procedures in order to match the behavior that United States customers exhibit most often. Data on freezing technique trends were accessible, but information on the actual operations carried out throughout these methods was missing, necessitating reliance on USDA recommendations and internal expert opinion [58]. In the absence of relevant *Campylobacter* transfer rate data, it was also assumed that *Campylobacter* transfer rates are comparable to those of the surrogates used in the included cross-contamination investigations. Other identified forms of cross-contamination events, like chicken washing and cross-contamination from other food products, were not included in the QMRA model [49]. For chicken washing, statistics on the transfer rate of chicken to different kitchen surfaces were unavailable. Cross-contamination from other foods was not considered since the present model was only focused on estimating the number of yearly illnesses caused by broiler meat. In addition, it was assumed that each customer ingested just one serving of chicken meat at a time, and their infection risk was calculated based on this single serving. In reality, people may take many portions in a single sitting, but are only infected once. Finally, this model was constructed with several varieties of broiler meat in mind, including chicken parts and chicken meal. While factors such as contamination, package size, consumption, and portion size may vary with different types of broiler meat, many of the parameters used in the QMRA model included data from numerous types of broiler meat; thus, a model with a broad scope was developed to estimate the risk posed by these various types of chicken meat. When further data become available, this model may be modified in the future to concentrate on a certain variety of chicken meat.

## 4. Conclusions

To conclude, the current QMRA model predicts the number of seasonal cases of campylobacteriosis caused by consuming chicken meat processed at home in the United States. There was a seasonal influence on the risk of infection per serving, with the risk of *Campylobacter* infection in chicken being lower during the winter months. Similarly, the frequency of infections and diseases was less during the winter than during other seasons. Comparing room-temperature thawing to alternative thawing procedures in a “what-if” scenario, the number of infections was much greater for the room-temperature thawing method. According to the results of the sensitivity analysis, the hand-washing, the transfer rate from hands to cooked chicken, and whether the hands are washed are the three most influential variables on the overall number of infections and illnesses. The model shows a framework for chicken consumption, from retail to preparation and consumption at home. It also points out research needs to make the predictions more accurate, as well as ways to reduce the risk of salmonellosis in the United States caused by eating chicken meat.

## Figures and Tables

**Figure 1 foods-12-02559-f001:**
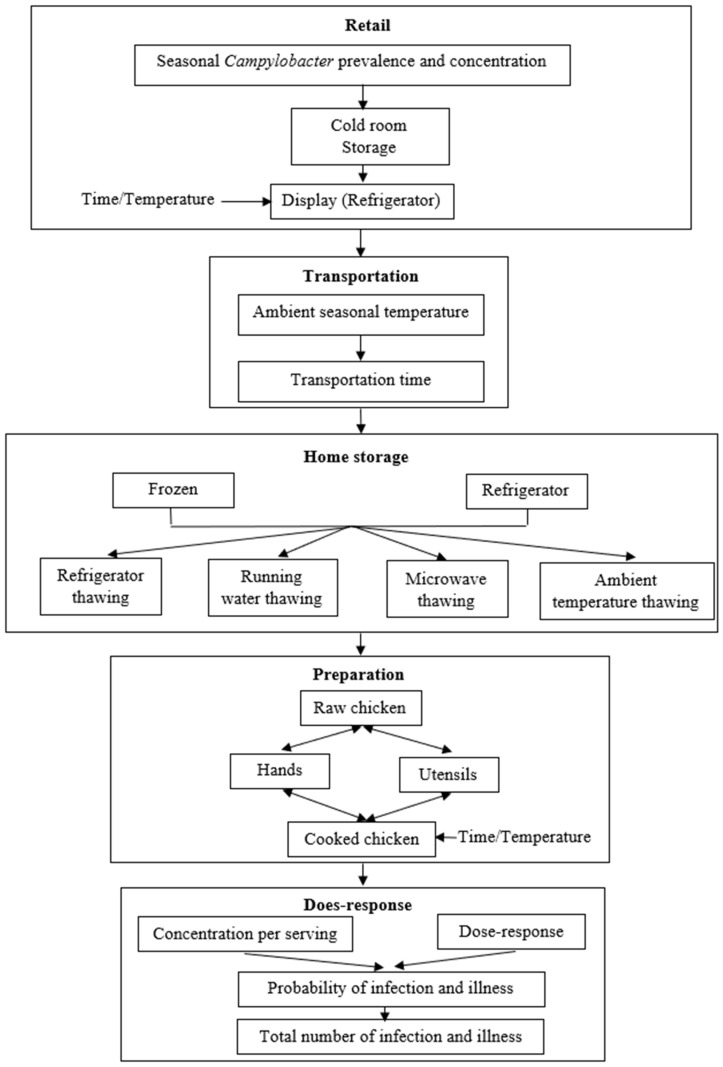
Overview of the quantitative microbial risk assessment model.

**Figure 2 foods-12-02559-f002:**
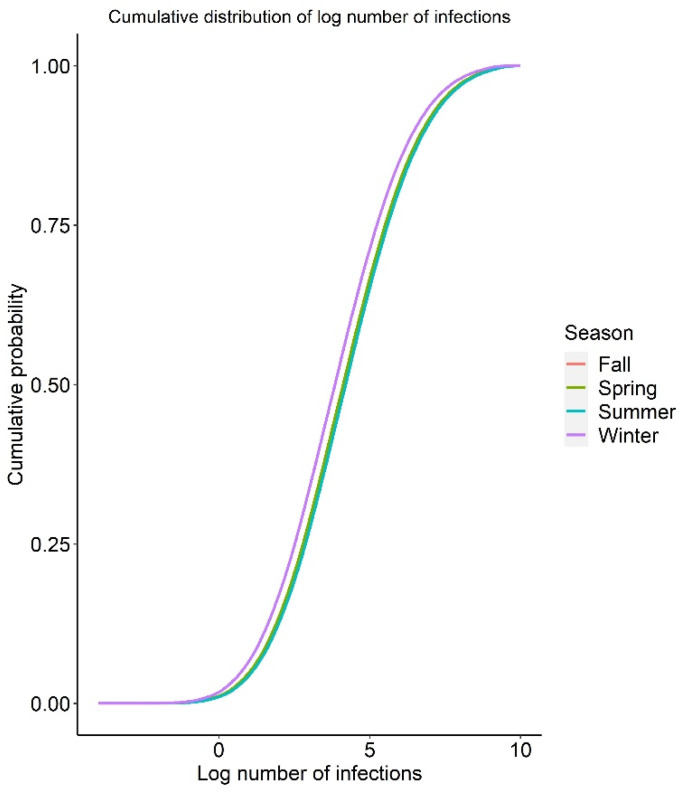
Cumulative distribution functions for log number of infections per season.

**Figure 3 foods-12-02559-f003:**
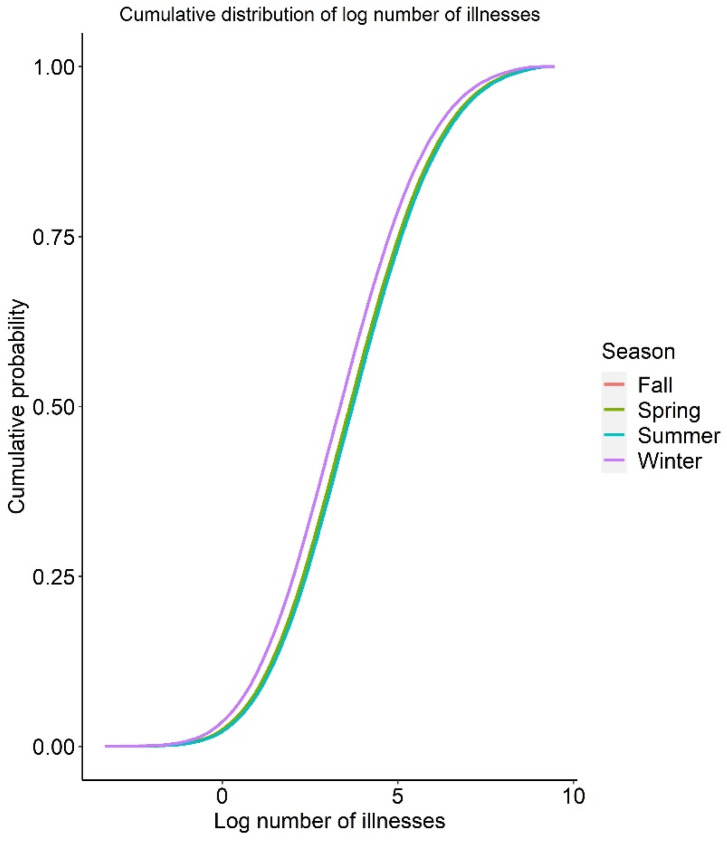
Cumulative distribution functions for log number of illnesses per season.

**Figure 4 foods-12-02559-f004:**
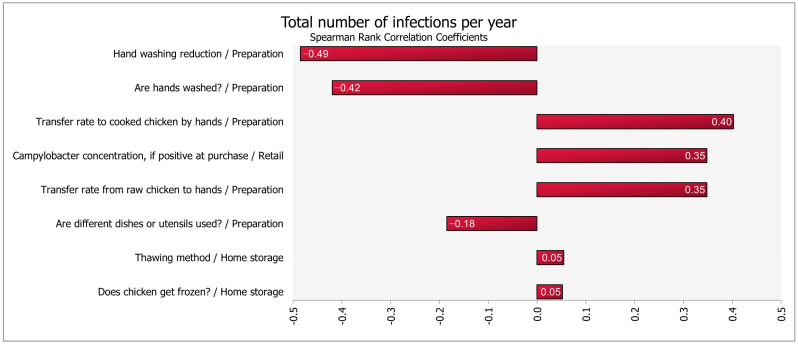
Spearman’s correlation coefficients show the eight most important model parameters for predicting the total number of infections in broiler meat.

**Table 1 foods-12-02559-t001:** Description of quantitative microbial risk assessment parameters in the baseline model.

Variable	Cell	Distribution, Value, or Formula	Unit	Source
**Growth Parameter**				
Growth model, b	B4	=0.04673	No unit	[22,23,24]
Growth model, T_min_	B5	=31.96	°C	[22,23,24]
Observed T_min_	B6	=31	°C	[25,26]
Newton heating constant, B	B7	=2.026	h^−1^	[27]
**Retail**				
Retail *Campylobacter* prevalence, Spring	B9	=RiskPert(0.41877,0.48933,0.48933)	Proportion	[10,28]
Retail *Campylobacter* prevalence, Summer	B10	=RiskTriang(0.546,0.546,0.608227)	Proportion	[10,28]
Retail *Campylobacter* prevalence, Fall	B11	=RiskUniform(0.507695,0.546205)	Proportion	[10,28]
Retail *Campylobacter* prevalence, Winter	B12	=RiskUniform(0.24443,0.268024)	Proportion	[10,28]
*Campylobacter* concentration, if positive at purchase	B13	=RiskWeibull(2.5448,1.9265,RiskShift(−1.4281))	log CFU/g	[29]
Retail cold room storage time	B14	=RiskExpon(0.58736,RiskShift(0.00027443)) × 4	h	[30]
Retail cold room storage temperature	B15	=RiskNormal(3.3188,1.7533)	°C	[30]
Retail display storage time	B16	=RiskExpon(0.22461,RiskShift(−0.0000889766)) × 24	h	[30]
Retail display storage temperature	B17	=RiskNormal(3.2321,1.3117)	°C	[30]
Growth rate during retail storage	B18	=IF(B15 < B6,0,(B4 × (B15 − B5))^2^) + IF(B17 < B6,0,(B4 × (B17 − B5))^2^)	log CFU/h	Calculated
Change during retail storage	B19	=B18 × (B14 + B16)	log CFU/g	Calculated
Concentration at point of purchase	B20	=IF((B19 + B13) > 5,5,B19 + B13)	log CFU/g	Calculated
**Transportation**				
Ambient temperature during transportation, Spring	B22	=RiskPert(1.7239,21.039,35.837)	°C	[31]
Ambient temperature during transportation, Summer	B23	=RiskPert(17.826,30.476,42.062)	°C	[31]
Ambient temperature during transportation, Fall	B24	=RiskPert(3.9191,21.889,39.226)	°C	[31]
Ambient temperature during transportation, Winter	B25	=RiskPert(−6.2993,10.263,29.747)	°C	[31]
Transportation time	B26	=RiskLoglogistic(0.0063772,1.0915,4.6212,RiskTruncate(0.3,18.45))	h	[32]
Transportation growth rate, Spring	B27	=IF(B22 < B6,0,(B4 × ((B22 − (EXP(−B7 × B26) × (B22 − B17))) − B5))^2^)	log CFU/h	Calculated
Transportation growth rate, Summer	B28	=IF(B23 < B6,0,(B4 × ((B23 − (EXP(−B7 × B26) × (B23 − B17))) − B5))^2^)	log CFU/h	Calculated
Transportation growth rate, Fall	B29	=IF(B24 < B6,0,(B4 × ((B24 − (EXP(−B7 × B26) × (B24 − B17))) − B5))^2^)	log CFU/h	Calculated
Transportation growth rate, Winter	B30	=IF(B25 < B6,0,(B4 × ((B25 − (EXP(−B7 × B26) × (B25 − B17))) − B5))^2^)	log CFU/h	Calculated
Change during transportation, Spring	B31	=B27 × B26	log CFU/g	Calculated
Change during transportation, Summer	B32	=B28 × B26	log CFU/g	Calculated
Change during transportation, Fall	B33	=B29 × B26	log CFU/g	Calculated
Change during transportation, Winter	B34	=B30 × B26	log CFU/g	Calculated
Concentration after transportation, Spring	B35	=B31 + B20	log CFU/g	Calculated
Concentration after transportation, Summer	B36	=B32 + B20	log CFU/g	Calculated
Concentration after transportation, Fall	B37	=B33 + B20	log CFU/g	Calculated
Concentration after transportation, Winter	B38	=B34 + B20	log CFU/g	Calculated
**Home storage**				
Does chicken get frozen?	B40	=RiskBernoulli(0.4)	No unit	[33]
If frozen:				
Time until frozen	B42	=RiskBetaGeneral(0.0067951,0.59992,0,2)	h	[33]
Ambient room temperature	B43	=RiskNormal(22.3107,5.8722,RiskTruncate(15,30))	°C	[34]
Growth rate before products were put in freezer	B44	=IF(B43 < B6,0,(B4 × (B43 − B5))^2^)	log CFU/h	Calculated
Change before frozen	B45	=B44 × B43	log CFU/g	Calculated
Concentration before frozen, Spring	B46	=B45 + B35	log CFU/g	Calculated
Concentration before frozen, Summer	B47	=B45 + B36	log CFU/g	Calculated
Concentration before frozen, Fall	B48	=B45 + B37	log CFU/g	Calculated
Concentration before frozen, Winter	B49	=B45 + B38	log CFU/g	Calculated
Home refrigerator temperature	B50	=RiskLaplace(4.4444,2.5231)	°C	[35]
Home freezer temperature	B51	=RiskNormal(−9.275,5.2857,RiskTruncate(−25,0))	°C	[36]
Thawing method	B52	=RiskDiscrete({1,2,3,4},{0.48,0.14,0.24,0.14})	No unit	[33]
If thaw method =1:				
Thaw time	B54	=RiskTriang(2,24,72)	h	
Growth rate during refrigerated thawing	B55	=IF(B50 < B6,0,(B4 × ((B50 − (EXP(−B7 × B54) × (B50 − B51))) − B5))^2^)	log CFU/h	Calculated
Change during refrigerated thawing	B56	=IF(B52 = 1,B54 × B55,0)	log CFU/g	Calculated
If thaw method = 2:				
Running water temperature	B58	=RiskPert(14,22.9,30)	°C	[37]
Thaw time	B59	=RiskTriang(0.25,1,2)		
Growth rate during running water thawing	B60	=(B4 × ((B58 − (EXP(−B7 × B59) × (B58 − B51))) − B5))^2^	log CFU/h	Calculated
Change during running water thawing	B61	=IF(B52 = 2,B59 × B60,0)	log CFU/g	Calculated
If thaw method = 3:				
Temperature of meat during microwave thawing	B63	=RiskPert(−8,−4,8)	°C	[38]
Thaw time	B64	=RiskUniform(8,20)/60	h	
Growth rate during microwave thawing	B65	=IF(B63 < B6,0,(B4 × ((B63 − (EXP(−B7 × B64) × (B63 − B51))) − B5))^2^)	log CFU/h	Calculated
Change during microwave thawing	B66	=IF(B52 = 3,B64 × B65,0)	log CFU/g	Calculated
If thaw method = 4:				
Ambient room temperature	B68	=RiskNormal(22.3107,5.8722,RiskTruncate(15,30))	°C	[34]
Thaw time	B69	=RiskUniform(1,10)	h	
Growth during room temperature thawing	B70	=(B4 × ((B68 − (EXP(−B7 × B69) × (B68 − B51))) − B5))^2^	log CFU/h	Calculated
Change during room temperature thawing	B71	=IF(B52 = 4,B69 × B70,0)	log CFU/g	Calculated
Concentration after thawing, Spring	B72	=IF(B40 = 1,B46 + B56 + B61 + B66 + B71,0)	log CFU/g	Calculated
Concentration after thawing, Summer	B73	=IF(B40 = 1,B47 + B56 + B61 + B66 + B71,0)	log CFU/g	Calculated
Concentration after thawing, Fall	B74	=IF(B40 = 1,B48 + B56 + B61 + B66 + B71,0)	log CFU/g	Calculated
Concentration after thawing, Winter	B75	=IF(B40 = 1,B49 + B56 + B61 + B66 + B71,0)	log CFU/g	Calculated
If not frozen:				
Refrigerator storage time	B78	=RiskPareto(3.4887,2,RiskTruncate(0,5)) × 24	h	[33]
Growth rate during refrigerated storage	B79	=IF(B50 < B6,0,(B4 × (B50 − B5))^2^)	log CFU/h	Calculated
Change during storage	B80	=B77 × B78	log CFU/g	Calculated
Concentration after storage, Spring	B81	=IF(B40 = 1,0,B35 + B79)	log CFU/g	Calculated
Concentration after storage, Summer	B82	=IF(B40 = 1,0,B36 + B79)	log CFU/g	Calculated
Concentration after storage, Fall	B83	=IF(B40 = 1,0,B37 + B79)	log CFU/g	Calculated
Concentration after storage, Winter	B84	=IF(B40 = 1,0,B38 + B79)	log CFU/g	Calculated
Concentration before preparation, Spring	B85	=B72 + B80	log CFU/g	Calculated
Concentration before preparation, Summer	B86	=B73 + B81	log CFU/g	Calculated
Concentration before preparation, Fall	B87	=B74 + B82	log CFU/g	Calculated
Concentration before preparation, Winter	B88	=B75 + B83	log CFU/g	Calculated
**Preparation**				
Raw chicken handling:				
Transfer rate from raw chicken to hands	B90	=RiskLognorm(0.15555,1.0547,RiskShift(0.00058696),RiskTruncate(0,1))	Proportion	[39,40,41]
Concentration on hands after handling, Spring	B91	=LOG10(B90 × (10^B85^))	log CFU/g	Calculated
Concentration on hands after handling, Summer	B92	=LOG10(B90 × (10^B86^))	log CFU/g	Calculated
Concentration on hands after handling, Fall	B93	=LOG10(B90 × (10^B87^))	log CFU/g	Calculated
Concentration on hands after handling, Winter	B94	=LOG10(B90 × (10^B88^))	log CFU/g	Calculated
Concentration left on chicken, Spring	B95	=IF(10^B85^–10^B91^ = 0,0,LOG10(10^B85^–10^B91^))	log CFU/g	Calculated
Concentration left on chicken, Summer	B96	=IF(10^B86^–10^B92^ = 0,0,LOG10(10^B86^–10^B92^))	log CFU/g	Calculated
Concentration left on chicken, Fall	B97	=IF(10^B87^–10^B93^ = 0,0,LOG10(10^B87^–10^B93^))	log CFU/g	Calculated
Concentration left on chicken, Winter	B98	=IF(10^B88^–10^B94^ = 0,0,LOG10(10^B88^–10^B94^))	log CFU/g	Calculated
Transfer rate from raw chicken to utensils	B99	=RiskLognorm(0.0064271,0.28575,RiskShift(0.00000124688),RiskTruncate(0,1))	Proportion	[39,40,41]
Concentration on utensils after handling, Spring	B100	=LOG10((10^B95^) × B99)	log CFU/g	Calculated
Concentration on utensils after handling, Summer	B101	=LOG10((10^B96^) × B99)	log CFU/g	Calculated
Concentration on utensils after handling, Fall	B102	=LOG10((10^B97^) × B99)	log CFU/g	Calculated
Concentration on utensils after handling, Winter	B103	=LOG10((10^B98^) × B99)	log CFU/g	Calculated
Concentration on chicken, Spring	B104	=LOG10(10^B95^–10^B100^)	log CFU/g	Calculated
Concentration on chicken, Summer	B105	=LOG10(10^B96^–10^B101^)	log CFU/g	Calculated
Concentration on chicken, Fall	B106	=LOG10(10^B97^–10^B102^)	log CFU/g	Calculated
Concentration on chicken, Winter	B107	=LOG10(10^B98^–10^B103^)	log CFU/g	Calculated
Cooking:				
Is chicken undercooked?	B109	=RiskBernoulli(0.399)	No unit	[32]
Cooking time	B110	=RiskPert(15,30,45,RiskCorrmat(NewMatrix1,1))	Min	[42]
Cooking temperature	B111	=RiskPert(38.244,82.305,100.48,RiskTruncate(38.244, 73.9),RiskCorrmat(NewMatrix1,2))	°C	[35]
D-value	B112	=10^(−0.96−(B111−70)/12.3)^	Min	[43]
Change during undercooking	B113	=B110/B112	log CFU/g	Calculated
Concentration after undercooking, Spring	B114	=B104 − B113	log CFU/g	Calculated
Concentration after undercooking, Summer	B115	=B105 − B113	log CFU/g	Calculated
Concentration after undercooking, Fall	B116	=B106 − B113	log CFU/g	Calculated
Concentration after undercooking, Winter	B117	=B107 − B113	log CFU/g	Calculated
Cooked product handling:				
Are hands washed?	B119	=RiskBernoulli(0.883)	No unit	[44]
Hand washing reduction	B120	=RiskNormal(2.7163,1.2661,RiskTruncate(0.34,5.29))	log CFU/g	[39]
Concentration on hands after washing, Spring	B121	=B91 − B120	log CFU/g	Calculated
Concentration on hands after washing, Summer	B122	=B92 − B120	log CFU/g	Calculated
Concentration on hands after washing, Fall	B123	=B93 − B120	log CFU/g	Calculated
Concentration on hands after washing, Winter	B124	=B94 − B120	log CFU/g	Calculated
Transfer rate to cooked chicken by hands	B125	=RiskLevy(−0.0003382,0.0019097,RiskTruncate(0,1))	Proportion	[39,40]
Concentration after handling cooked chicken with hands, Spring	B126	=LOG10(IF(B119 = 0,(10^B91^) × B125,(10^B121^) × B125) + IF(B109 = 0,0, 10^B114^))	log CFU/g	Calculated
Concentration after handling cooked chicken with hands, Summer	B127	=LOG10(IF(B119 = 0,(10^B92^) × B125,(10^B122^) × B125) + IF(B109 = 0,0, 10^B115^))	log CFU/g	Calculated
Concentration after handling cooked chicken with hands, Fall	B128	=LOG10(IF(B119 = 0,(10^B93^) × B125,(10^B123^) × B125) + IF(B109 = 0,0, 10^B116^))	log CFU/g	Calculated
Concentration after handling cooked chicken with hands, Winter	B129	=LOG10(IF(B119 = 0,(10^B94^) × B125,(10^B124^) × B125) + IF(B109 = 0,0, 10^B117^))	log CFU/g	Calculated
Are different dishes or utensils used?	B130	=RiskBernoulli(0.959)	No unit	[44]
Transfer rate to cooked chicken by dirty utensils	B131	=RiskExpon(0.12217,RiskShift(−0.00041787),RiskTruncate(0,1))	Proportion	[39]
Final concentration, Spring	B132	=LOG10(10^B126^ + IF(B130 = 0,B131 × (10^B100^),0))	log CFU/g	Calculated
Final concentration, Summer	B133	=LOG10(10^B127^ + IF(B130 = 0,B131 × (10^B101^),0))	log CFU/g	Calculated
Final concentration, Fall	B134	=LOG10(10^B128^ + IF(B130 = 0,B131 × (10^B102^),0))	log CFU/g	Calculated
Final concentration, Winter	B135	=LOG10(10^B129 + IF(B130 = 0,B131 × (10^B103^),0))	log CFU/g	Calculated
**Dose–response and infection**				
Serving size	B137	=85	g	9 CFR §381.412
Concentration per serving, Spring	B138	=(10^B132^) × B137	CFU	Calculated
Concentration per serving, Summer	B139	=(10^B133^) × B137	CFU	Calculated
Concentration per serving, Fall	B140	=(10^B134^) × B137	CFU	Calculated
Concentration per serving, Winter	B141	=(10^B135^) × B137	CFU	Calculated
Dose–response infection model parameter alpha	B142	=0.145	No unit	[45]
Dose–response infection model parameter, beta	B143	=7.59	No unit	[45]
Probability of infection, Spring	B144	=1 − (1 + (B138/B143))^−B142^	No unit	Calculated
Probability of infection, Summer	B145	=1 − (1 + (B139/B143))^−B142^	No unit	Calculated
Probability of infection, Fall	B146	=1 − (1 + (B140/B143))^−B142^	No unit	Calculated
Probability of infection, Winter	B147	=1 − (1 + (B141/B143))^−B142^	CFU	Calculated
Probability of illness, Spring	B148	=B144 × 0.33	No unit	[46,47,48]
Probability of illness, Summer	B149	=B145 × 0.33	No unit	[46,47,48]
Probability of illness, Fall	B150	=B146 × 0.33	No unit	[46,47,48]
Probability of illness, Winter	B151	=B147 × 0.33	No unit	[46,47,48]
Risk of infection per serving, Spring	B152	=B144 × B9	No unit	Calculated
Risk of infection per serving, Summer	B153	=B145 × B10	No unit	Calculated
Risk of infection per serving, Fall	B154	=B146 × B11	No unit	Calculated
Risk of infection per serving, Winter	B155	=B147 × B12	No unit	Calculated
Risk of illness per serving, Spring	B156	=B148 × B9	No unit	Calculated
Risk of illness per serving, Summer	B157	=B149 × B10	No unit	Calculated
Risk of illness per serving, Fall	B158	=B150 × B11	No unit	Calculated
Risk of illness per serving, Winter	B159	=B151 × B12	No unit	Calculated
Total per capita poultry availability per year	B160	=43,454.15	g	[49]
Total used in raw chicken preparation per year	B161	=21,727.075	g	[50]
U.S. population	B162	=325,186,237	People	[49]
Number of consumers who purchased chicken from grocery/supermarket	B163	=269,904,576.7	People	[51]
Consumed serving per person per season	B164	=63.90	Serving	Calculated
No. of servings consumed per season in US	B165	=17,247,755,827	No unit	Calculated
No. of infections per season, Spring	B166	=B165 × B152	No unit	Calculated
No. of infections per season, Summer	B167	=B165 × B153	No unit	Calculated
No. of infections per season, Fall	B168	=B165 × B154	No unit	Calculated
No. of infections per season, Winter	B169	=B165 × B155	No unit	Calculated
No. of illness per season, Spring	B170	=B165 × B156	No unit	Calculated
No. of illness per season, Summer	B171	=B165 × B157	No unit	Calculated
No. of illness per season, Fall	B172	=B165 × B158	No unit	Calculated
No. of illness per season, Winter	B173	=B165 × B159	No unit	Calculated
Total number of infections per year	B174	=B166 + B167 + B168 + B169	No unit	Calculated
Total number of illnesses per year	B175	=B170 + B171 + B172 + B173	No unit	Calculated

**Table 2 foods-12-02559-t002:** Seasonal trends of *Campylobacter* prevalence, concentrations, and outbreaks in chicken products.

	Spring	Summer	Fall	Winter
*Campylobacter* prevalence (Average ± SD)	0.59 ± 0.32	0.56 ± 0.48	0.53 ± 0.41	0.26 ± 0.32
*Campylobacter* concentration (log CFU/carcass) (Average ± SD)	2.26 ± 0.56	1.74 ± 0.89	2.35 ± 0.85	2.30 ± 1.28
*Campylobacter* outbreaks *	17	25	15	10

* Outbreaks data were extracted from the National Outbreak Reporting System (NORS) from 1998 to 2020 and are strictly related to chicken and *Campylobacter*.

**Table 3 foods-12-02559-t003:** Summary statistics of risk of infection and illness per season determined by QMRA baseline model.

Seasonal Effect	Risk of Infection per Serving				Risk of Illness per Serving			
	Mean	Median	25%	75%	Mean	Median	25%	75%
Spring	$1.31\times 10^{-3}$	$7.57\times 10^{-7}$	$3.56\times 10^{-8}$	$1.85\times 10^{-5}$	$4.32\times 10^{-4}$	$2.50\times 10^{-7}$	$1.17\times 10^{-8}$	$6.09\times 10^{-6}$
Summer	$1.57\times 10^{-3}$	$9.22\times 10^{-7}$	$4.29\times 10^{-8}$	$2.24\times 10^{-5}$	$5.18\times 10^{-4}$	$3.04\times 10^{-7}$	$1.42\times 10^{-8}$	$7.40\times 10^{-6}$
Fall	$1.45\times 10^{-3}$	$8.40\times 10^{-7}$	$3.92\times 10^{-8}$	$2.05\times 10^{-5}$	$4.77\times 10^{-4}$	$2.77\times 10^{-7}$	$1.29\times 10^{-8}$	$6.75\times 10^{-6}$
Winter	$7.01\times 10^{-4}$	$4.06\times 10^{-7}$	$1.89\times 10^{-8}$	$9.91\times 10^{-6}$	$2.31\times 10^{-4}$	$1.34\times 10^{-7}$	$6.25\times 10^{-9}$	$3.27\times 10^{-6}$

**Table 4 foods-12-02559-t004:** Summary statistics of number of infections and illnesses per season.

Seasonal Effect	No. of Infections per Season				No. of Illnesses per Season			
	Mean	Median	25%	75%	Mean	Median	25%	75%
Spring	22,571,609	13,050	611	318,258	7,448,639	4306	201	105,034
Summer	27,058,680	15,895	739	386,805	8,929,364	5245	244	127,646
Fall	24,941,190	14,488	676	353,010	8,230,593	4781	223	116,493
Winter	12,085,638	7008	327	170,841	3,988,261	2312	108	56,378

**Table 5 foods-12-02559-t005:** Summary statistics for total number of infections annually of uncertainty analysis.

Scenario	No. of Infections			
	Mean	Median	25%	75%
Baseline	86,657,118	50,493	2355	1,228,846
*Uncertainty, prevalence:*				
Low	79,734,367	48,471	2179	1,212,001
Medium	90,535,132	50,282	2375	1,305,955
High	92,231,575	53,480	2525	1,349,224
*Thawing method:*				
Refrigerator thawing	53,162,035	42,674	2014	1,010,711
Running water thawing	68,289,580	61,462	2887	1,487,499
Microwave thawing	53,007,197	42,047	2018	1,014,576
Ambient room temperature thawing	286,663,540	122,142	4714	3933
*Hand washing:*				
Always wash hands	41,774,442	26,583	1569	471,588
Never wash hands	429,585,788	11,308,668	1,386,150	100,598,358
*Cleaning:*				
Always use different utensils	83,552,680	42,190	2046	1,027,979
Never use different utensils	213,628,883	2,694,612	312,481	25,216,122

## Data Availability

The data used to support the findings of this study can be made available by the corresponding author upon request.

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
