# Peer review of "Assessing the Risk of Seasonal Effects of Campylobacter Contaminated Broiler Meat Prepared In-Home in the United States"

_foods, 2023, doi:10.3390/foods12132559_

Round 1
Reviewer 1 Report
Dear Authors,
Your manuscript has been reviewed,
This work deserves attention, since it highlights an important topic related to the Assessment of the risk of having campylobacteriosis due to contaminated broiler meat prepared in-home in the United States in different seasons.
The paper is well written in English language, well presented and the design is good. Kindly find below my comments and suggestions regarding your work.
01- In the whole manuscript, Authors are invited to delete the empty spaces, example are Lines 26, 27 and 28.
02- In this study, I have a question concerning the purpose of the title and the whole manuscript, why authors were focusing on the effect of seasonality and not on the monthly temperature during the whole year? Noting that some Fall months can be warmer than Summer ones, and so on.
03- In the Introduction section, Authors are invited to start this part by a small introduction about campylobacteriosis worldwide, in developed and developing countries, in children and adults, as one of main causes of bacterial gastroenteritis. They can use these references for this point:
Ref 01: Global Epidemiology of Campylobacter Infection
Ref 02: Risk factors for indigenous Campylobacter jejuni and Campylobacter coli infections in The Netherlands: a case-control study
04- In the Introduction section, when authors talk about the prevalence of Campylobacter in broiler, authors are invited to use this reference:
Ref 01: Prevalence of Campylobacter spp. in broilers in North Lebanon
05- In the Introduction section, when authors talk about Campylobacter and seasonality (Line 51) they are invited to read and use as reference the following article which represent a very important one related to this point:
Ref 01: Prevalence, antimicrobial resistance and risk factors for campylobacteriosis in Lebanon
06- In the Introduction section, Line 56, after the term consumers, Authors are invited to add a reference for this point.
Best Regards,
Author Response
We thank the reviewer for their positive and constructive feedback.
Dear Authors,
Your manuscript has been reviewed,
This work deserves attention, since it highlights an important topic related to the Assessment of the risk of having campylobacteriosis due to contaminated broiler meat prepared in-home in the United States in different seasons.
The paper is well written in English language, well presented, and the design is good. Kindly find below my comments and suggestions regarding your work.
01- In the whole manuscript, Authors are invited to delete the empty spaces; examples are Lines 26, 27, and 28.
AU: Deleted as suggested.
02- In this study, I have a question concerning the purpose of the title and the whole manuscript; why authors were focusing on the effect of seasonality and not on the monthly temperature during the whole year? Noting that some Fall months can be warmer than summer ones, and so on.
AU: The focus on seasonality instead of monthly temperature in the research is due to the following reasons:
- Simplicity: It's easier to analyze data and create models based on four seasonal classifications than twelve monthly classifications. More categories increase complexity, require more data, and make interpretation and communication of results more challenging.
- Generalizability: The season-based model may be more applicable and generalizable to a broader context. For example, the risk associated with consuming broiler meat in the summer is high regardless of the specific month.
- Seasonal Behaviors: The study also considers changes in human behavior and agricultural practices by season. For instance, people may consume more barbecued chicken in the summer, or farming practices might change by season.
- Climatic Variations: Even though some Fall months can be warmer than summer ones, the overall climate during a season (including factors like humidity and precipitation) might affect the survival and multiplication of Campylobacter.
Your point is very valid. A more subtle model that considers monthly variations in temperature (or even weekly or daily variations) could potentially provide a more accurate risk assessment. This could be a direction for future research.
03- In the Introduction section, Authors are invited to start this part by a small introduction about campylobacteriosis worldwide, in developed and developing countries, in children and adults, as one of main causes of bacterial gastroenteritis. They can use these references for this point:
Ref 01: Global Epidemiology of Campylobacter Infection
Ref 02: Risk factors for indigenous Campylobacter jejuni and Campylobacter coli infections in The Netherlands: a case-control study
AU: The small introduction is added with suggested references.
Campylobacteriosis, a foodborne illness caused by bacteria of the genus Campylo-bacter, poses a significant global health burden. This bacterial infection is one of the leading causes of gastroenteritis worldwide, with millions of people affected annually (Kaakoush et al., 2014). Affecting both developing and developed nations, the disease is primarily linked to the consumption of contaminated poultry, unpasteurized milk, and untreated water. In developed countries, campylobacteriosis is often a sporadic, rather than epidemic, issue, with cases peaking during the warmer months (Doorduyn et al., 2010). Conversely, in developing countries, Campylobacter infections are often endemic and more frequently affect children under the age of five (Kirk et al., 2015).
04- In the Introduction section, when authors talk about the prevalence of Campylobacter in broiler, authors are invited to use this reference:
Ref 01: Prevalence of Campylobacter spp. in broilers in North Lebanon
AU: The following sentence is added as suggested.
A study revealed the prevalence of Campylobacter spp. in broilers from North Lebanon, with high infection rates in both open rearing system (92%) and closed rearing system (85%) (Awada et al., 2023).
05- In the Introduction section, when authors talk about Campylobacter and seasonality (Line 51) they are invited to read and use as reference the following article which represent a very important one related to this point:
Ref 01: Prevalence, antimicrobial resistance and risk factors for campylobacteriosis in Lebanon
AU: The following sentence is added as suggested.
In a study carried out in Lebanon also reported the highest Campylobacter infection rates in summer (31.6%) (Ibrahim et al., 2019).
06- In the Introduction section, Line 56, after the term consumers, Authors are invited to add a reference for this point.
AU: The following references are added to support the point.
Dogan, O. B., Clarke, J., Mattos, F., & Wang, B. (2019). A quantitative microbial risk assessment model of Campylobacter in broiler chickens: Evaluating processing interventions. Food Control, 100, 97-110.
Membré, J. M., & Boué, G. (2018). Quantitative microbiological risk assessment in food industry: Theory and practical application. Food Research International, 106, 1132-1139.

Reviewer 2 Report
Peer-review of the manuscript "Assessing the risk of seasonal effects of Campylobacter contaminated broiler meat prepared in-home in the United States"
The study is a mathematical model on the microbial risk assessment of chicken meat consumption in the United States with respect to the pathogen Campylobacter spp. The main conclusion of the study is the difference in the risk of infection in the summer season compared to the winter months. However, in other conclusions, the study does not bring anything new. The need to wash hands when handling raw meat to prevent cross-contamination is generally known. Likewise, the necessity to use clean utensils when preparing raw meat in the kitchen.
On the contrary, the study has several shortcomings that need to be corrected:
1) The study does not address the issue of chicken meat packaging in retail at all. What effect does a modified atmosphere packaging or vacuum packaging have on the survival of Campylobacter spp.? Campylobacter is a microaerophilic bacterium requiring an atmosphere of 3-15% oxygen and 2-10% carbon dioxide. How will an atmosphere with 20-30% carbon dioxide or a vacuum affect its survival?
2) Campylobacter is highly sensitive to oxygen and does not survive in an aerobic atmosphere. If the authors address the issue of freezing chicken meat in consumer households, is the meat packaged or not? Chicken meat taken out of the package is exposed to the air and under these conditions campylobacter cannot multiply even at a temperature of 31 °C and above. This should be mentioned in the study.
3) The study uses incorrect terminology in many places. The term "concentration" is used to express the content of chemical substances. In the case of bacteria, I recommend using the term “contamination level”. On line 199, the term “infected utensils” must be replaced by the term “contaminated utensils”, as well as on line 231 “chicken meat infected with Campylobacter” should correctly be “chicken meat contaminated with Campylobacter”. People or animals are infected, food or other inanimate things are contaminated.
4) Lines 280 and 281 show 0.41 percent of contaminated skinless chicken breast cuts? Shouldn't it be 41%?
5) In tab 2, the statistical significance (P value) of the differences in the results should be indicated.
6) I recommend explaining the huge differences between means and medians in tab. 3 and 4.
7) I appreciate that all cited articles and sources in the References chapter are in-text. However, on line 464 the Montville et al. 2002 missing from References.
8) The names of some journals are incorrectly listed in the References chapter - words should be capitalized in the titles: Food Microbiology; BMC Microbiology etc.
9) Tab. 1 is excessively extensive. I recommend publishing it in its entirety in the form of a Supplementary File and placing its brief form in the article on a maximum of 1 page of text.
1The study uses incorrect terminology in many places. The term "concentration" is used to express the content of chemical substances. In the case of bacteria, I recommend using the term “contamination level”. On line 199, the term “infected utensils” must be replaced by the term “contaminated utensils”, as well as on line 231 “chicken meat infected with Campylobacter” should correctly be “chicken meat contaminated with Campylobacter”. People or animals are infected, food or other inanimate things are contaminated.
Author Response
We thank the reviewer for their time and constructive feedback.
Response to Reviewer 2 Comments
Peer-review of the manuscript "Assessing the risk of seasonal effects of Campylobacter contaminated broiler meat prepared in-home in the United States"
The study is a mathematical model on the microbial risk assessment of chicken meat consumption in the United States with respect to the pathogen Campylobacter spp. The main conclusion of the study is the difference in the risk of infection in the summer season compared to the winter months. However, in other conclusions, the study does not bring anything new. The need to wash hands when handling raw meat to prevent cross-contamination is generally known. Likewise, the necessity to use clean utensils when preparing raw meat in the kitchen.
On the contrary, the study has several shortcomings that need to be corrected:
- The study does not address the issue of chicken meat packaging in retail at all. What effect does a modified atmosphere packaging or vacuum packaging have on the survival of Campylobacter? Campylobacter is a microaerophilic bacterium requiring an atmosphere of 3-15% oxygen and 2-10% carbon dioxide. How will an atmosphere with 20-30% carbon dioxide or a vacuum affect its survival?
AU: Modified atmosphere packaging (MAP) and vacuum packaging can extend shelf life by creating conditions unfavorable for the growth of spoilage and pathogenic bacteria. MAP involves replacing the air inside the packaging with a mixture of gases, often including nitrogen, oxygen, and carbon dioxide. High levels of carbon dioxide (20-30%) could inhibit the growth of Campylobacter spp. due to their sensitivity to high CO2 levels. Carbon dioxide has antimicrobial properties, including the ability to permeate the cell wall of bacteria, disrupt metabolism, and inhibit growth. Vacuum packaging involves removing all the air (and therefore oxygen) from the packaging before sealing it. As a microaerophilic bacterium, Campylobacter spp. could survive in low-oxygen environments, but the absence of oxygen in vacuum packaging may be too extreme, and potentially limit their survival or growth. Though MAP and vacuum packaging may have an inhibitory effect on microorganisms, the purpose of the study is to assess the risk of Campylobacter in broiler meat from the farm to retail and home storage, focusing on variables such as temperature and time, rather than packaging methods. Nevertheless, to compare the effect of packagings (Foam trays with plastic overwrap, MAP, vacuum packaging) on Campylobacter growth and survival is of great importance and interest for future studies.
- Campylobacter is highly sensitive to oxygen and does not survive in an aerobic atmosphere. If the authors address the issue of freezing chicken meat in consumer households, is the meat packaged or not? Chicken meat taken out of the package is exposed to the air and under these conditions campylobacter cannot multiply even at a temperature of 31 °C and above. This should be mentioned in the study.
AU: We assume the meat is in its original packaging when put into freezer. So as when consumer take it from freezer to thaw. Though Campylobacter are microaerophilic bacteria, it can survive under aerobic conditions (Bronowski et al., 2014; Rodrigues et al., 2020; Smith et al., 2016). The survival of Campylobacter is well studied, many more research have shown that Campylobacter is capable of adapting to aerobic environment.
Bronowski, C., James, C. E., & Winstanley, C. (2014). Role of environmental survival in transmission of Campylobacter jejuni. FEMS microbiology letters, 356(1), 8-19.
Smith, S., Meade, J., Gibbons, J., McGill, K., Bolton, D., & Whyte, P. (2016). The impact of environmental conditions on Campylobacter jejuni survival in broiler faeces and litter. Infection ecology & epidemiology, 6(1), 31685.
Rodrigues, R. C., Pocheron, A. L., Hernould, M., Haddad, N., Tresse, O., & Cappelier, J. M. (2015). Description of Campylobacter jejuni Bf, an atypical aero-tolerant strain. Gut pathogens, 7, 1-12.
- The study uses incorrect terminology in many places. The term "concentration" is used to express the content of chemical substances. In the case of bacteria, I recommend using the term “contamination level”. On line 199, the term “infected utensils” must be replaced by the term “contaminated utensils”, as well as on line 231 “chicken meat infected with Campylobacter” should correctly be “chicken meat contaminated with Campylobacter”. People or animals are infected, food or other inanimate things are contaminated.
AU: Concentration is also a common way of expressing bacteria population.
On line 199, the term is revised as suggested. Below is the sentence.
In the baseline QMRA model, the following cross-contamination scenarios were evaluated: raw chicken to hands, raw chicken to utensils (e.g., cutting boards, knives, etc.), hands to cooked chicken, and contaminated chicken.
On line 231, the term is revised as suggested. Below is the sentence.
As a final result, we intend to estimate the probability of infection and illness resulting from consuming chicken meat contaminated and infected with Campylobacter.
The usage of ‘infect’ is checked throughout the manuscript to ensure it is correctly used.
4) Lines 280 and 281 show 0.41 percent of contaminated skinless chicken breast cuts. Shouldn't it be 41%?
AU: It should be 41 percent. The sentence is revised as below.
In Alabama, 0.41 percent of skinless chicken breasts were contaminated with Campylobacter (Williams and Oyarzabal, 2012).
5) In tab 2, the statistical significance (P value) of the differences in the results should be indicated.
AU: Table 2 shows the mean and standard deviation of Campylobacter prevalence and concentration. In calculation of mean and standard deviation, p value is not involved.
6) I recommend explaining the huge differences between means and medians in tab. 3 and 4.
AU: When the mean is much larger than the median in a quantitative microbial risk assessment (QMRA), it suggests that the data is positively skewed. This is quite common in risk assessment data due to the nature of the factors being measured. A risk of infection is often a small probability, and for most people, the risk might be very low. However, there might be a few individuals or scenarios where the risk of infection is extremely high. This creates a long tail on the right side of the distribution, leading to a positive skew.
In this case, the median (the middle value) is less affected by these high-risk scenarios and thus provides a better representation of the "typical" risk of infection. On the other hand, the mean (the average) is influenced by all values, including the high-risk ones, making it much larger than the median in a positively skewed distribution.
7) I appreciate that all cited articles and sources in the References chapter are in-text. However, on line 464 the Montville et al. 2002 missing from References.
AU: The missing reference is added.
8) The names of some journals are incorrectly listed in the References chapter - words should be capitalized in the titles: Food Microbiology; BMC Microbiology etc.
AU: The names of the journals are revised.
9) Tab. 1 is excessively extensive. I recommend publishing it in its entirety in the form of a Supplementary File and placing its brief form in the article on a maximum of 1 page of text.
AU: Table 1 is the most important table showing the parameters used in the QMAR model. With the complete table, others can reproduce our results. Also, in method and result section, the manuscript refer to the table heavily, without showing it completely it may cause confusions to readers. I highly recommend put the entire table in the main content.
Reference:
Bronowski, C., James, C. E., & Winstanley, C. (2014). Role of environmental survival in transmission of Campylobacter jejuni. FEMS Microbiol Lett, 356(1), 8-19. https://doi.org/10.1111/1574-6968.12488
Rodrigues, D. R., Briggs, W., Duff, A., Chasser, K., Murugesan, R., Pender, C., Ramirez, S., Valenzuela, L., & Bielke, L. (2020). Cecal microbiome composition and metabolic function in probiotic treated broilers. PLoS One, 15(6), e0225921. https://doi.org/10.1371/journal.pone.0225921
Smith, S., Meade, J., Gibbons, J., McGill, K., Bolton, D., & Whyte, P. (2016). The impact of environmental conditions on Campylobacter jejuni survival in broiler faeces and litter. Infect Ecol Epidemiol, 6, 31685. https://doi.org/10.3402/iee.v6.31685

Round 2
Reviewer 1 Report
Dear Authors,
You present manuscript is better in its form than the previous one.
Thank you for the modifications you did,
Best Regards,